# Deep Bayesian Experimental Design for Quantum Many-Body Systems

**Leopoldo Sarra**
Max Planck Institute for the Science of Light
Erlangen, 91058 Germany
`leopoldo.sarra@mpl.mpg.de`

**Florian Marquardt**
Max Planck Institute for the Science of Light
Erlangen, 91058 Germany
`florian.marquardt@mpl.mpg.de`

## Abstract

Bayesian experimental design is a technique that allows to efficiently select measurements to characterize a physical system by maximizing the expected information gain. Recent developments in deep neural networks and normalizing flows allow for a more efficient approximation of the posterior and thus the extension of this technique to complex high-dimensional situations. In this paper, we show how this approach holds promise for adaptive measurement strategies to characterize present-day quantum technology platforms. In particular, we focus on arrays of coupled cavities and qubit arrays. Both represent model systems of high relevance for modern applications, like quantum simulations and computing, and both have been realized in platforms where measurement and control can be exploited to characterize and counteract unavoidable disorder. Thus, they represent ideal targets for applications of Bayesian experimental design. *To reference this work, please cite the original journal article where this manuscript was first published: L. Sarra and F. Marquardt 2023, Mach. Learn.: Sci. Technol. 4, 045022 (2023).*[1]

## 1 Introduction

We are currently witnessing rapid scaling in the number of components for quantum technology platforms. Fulfilling the promise of fault-tolerant quantum computation will eventually require millions of qubits, and while current implementations still fall far short of that goal, the specific challenges of scaling are already apparent for the present-day devices including on the order of hundred qubits [26]. Other areas like integrated photonic circuits [52] are following a similar trajectory, for applications such as neuromorphic computing [53] or sensing and more fundamental studies in areas like topological transport [40]. There, large networks of beamsplitters, waveguides and resonators, again with component counts on the order of dozens or hundreds, are being fabricated and deployed. Similar large-scale networks have now been fabricated and investigated for coupled mechanical resonators, producing phononic circuits with local access to vibrational modes.

Characterizing any of these devices is a very important but nontrivial task, especially if it is to be done in an efficient manner, within a limited time budget [20]. Active learning [47], in the form of optimal experimental design can help, provided that one can employ techniques able to deal with the large number of parameters that are to be determined. These parameters comprise resonance

---

[1]"This extended abstract is derived from the accepted manuscript "Deep Bayesian experimental design for quantum many-body systems", before editing by Machine Learning: Science and Technology. IOP Publishing Ltd is not responsible for any errors or omissions in this version of the manuscript or any version derived from it. The Version of Record is available online at 10.1088/2632-2153/ad020d [46].

37th Conference on Neural Information Processing Systems (NeurIPS 2023).

frequencies of optical or microwave modes, couplings between components (e.g. between resonators and qubits or waveguides, or beamsplitter transparencies), nonlinearities, propagation phase shifts, matrix elements for the effect of external drives, and decay rates. Many measurement approaches can be drawn upon, each of them coming with its respective parameters that can be adjusted prior to each new measurement. In linear devices, specific components of the scattering matrix can be measured by injecting waves in some port and performing a homodyne measurement on another port. Here, the frequency would be a continuous parameter, while, depending on the setup, the choice of ports could be another, discrete parameter. In nonlinear systems, such as circuits comprised of qubits, manipulation via pulses in Ramsey-type schemes is the natural choice, with a final projective measurement of one or several qubits. The drive amplitude and pulse times would then represent the measurement parameters to be optimized. Systems that couple qubits and resonators or waveguides can also be characterized via nonlinear transmission, with the amplitude and drive frequency being treated as adjustable.

In all these cases, there is unavoidable fundamental noise in the measurement outcomes, namely shot noise in the case of wave transmission measurements and quantum projection noise in the case of qubit measurements [8]. This noise can naturally be reduced by extending the measurement duration, increasing the wave amplitude, or repeating multiple times the qubit pulse sequence together with its final projective qubit measurement (multi-shot measurement). As a consequence, the information gain per such extended measurement increases. However, there are limits to the usefulness of this naive optimization: wave amplitudes can be increased only so far before entering nonlinear regimes or heating up the device. In addition, one cannot keep measuring at only one single choice of the measurement parameter, since that will eventually only pinpoint a particular function of the many underlying setup parameters and not allow to resolve all the parameters individually. That is where active learning, i.e. choosing next measurement settings efficiently, can offer true benefits.

In this paper, we investigate recent and promising techniques from machine learning [16, 27] to efficiently and accurately make the best parameter prediction given past measurement, and propose the next best measurement to perform. We show applications of these techniques to the above-mentioned quantum devices. We focus especially on quantum systems, such as chains of coupled cavities and arrays of qubits. The presented framework is nonetheless completely general and appliable in a similar way to other settings.

## 2  Related Work

The idea of finding the best experiment to perform is known as Active Learning in the machine learning literature [10, 47, 49] and as Bayesian optimal experimental design [6] in more specific parameter estimation applications [18, 43]. It is very common in science to have a class of possible models, dependent on a set of parameters, and to perform experiments to find those that better describe the true system. The outcome of each experiment does not give direct knowledge on the parameters themselves because of noise and measurement errors, but it provides partial information. Therefore, it is generally not enough to perform as many measurements as the number of unknown parameters of the system, and more experiments are required. When enough information is collected through measurements, we can identify the true parameters with a certain confidence.

When experiments are expensive, either in terms of cost, time and effort, it can be important to be efficient in the number of experiments required to characterize the given physical system. In those cases, it is desirable to exploit the information obtained from each experiment as much as possible, and to choose the sequence of experiments in such a way that the smallest number of experiments is required.

Information Theory provides satisfactory answers to this problem, at least from the theoretical point of view. In particular, the Bayes theorem shows the proper way to include a new measurement into our knowledge and update our predictions of the parameters of the system, and expected information gain [35] is the quantity to look at to find the best possible experiments to perform.

There have already been many remarkable applications in science [19, 44] and in physics in particular, ranging from devising quantum metrological procedures [24] to characterizing quantum dots [33, 39], superconducting quantum processors [22], or sensors [14], multi-phase estimation [50, 7], and their use is becoming more and more common. Parameter estimation can then be further generalized to

the complete design of the experiment. For example, [31, 38, 30] completely automate the search for the experiment that solves a given task, in the case of quantum optics.

Nevertheless, optimal experimental design techniques have not been used in their full power up to recently, even though these techniques have been known for decades. The main reason is that the exact evaluation of those statistical quantities can be very expensive. Therefore, very rough approximations were usually made, including the use of empirical heuristics, the use of Gaussian processes [34] and Gaussian posterior approximation [51, 13] and the optimization through maximum likelihood estimation [32]. The approximation usually depends on where the bottleneck is in practice, according to the specific problem: high-dimensional parameter space, large number of different possible experiments to perform, large number of measured quantities of each experiment, experiment execution time and cost, and total allowed number of experiments.

Modern neural networks [21] that employ normalizing flows [12, 29] for the approximation of the posterior distribution, developed in the past few years, recently allowed for quite more efficient estimations [16, 27] that do not require such rough (and often unjustified) assumptions. The price to pay for the increased precision of the approximation is clearly a larger computational effort.

## 3   Methods

Bayesian experimental design can be implemented as follows. The settings of the experiment, which could be, for example, a measurement at a particular frequency, are described by a vector $x$. The physical system is identified by hidden parameters $\lambda$, which we would like to determine. Because of measurement noise, the outcome of the experiment, $y$, is not deterministic, but distributed according to a distribution $P(y|\lambda, x)$, called likelihood of an observation. We can imagine, for example, $y = f(x, \lambda) + \epsilon$, where $f$ is a deterministic function and $\epsilon$ is a random variable, e.g. Gaussian distributed. We can update our knowledge of the parameters of the system with the Bayes rule [42]: given our inferred distribution $P_n(\lambda|\mathcal{M}_n)$ after $n$ measurements $\mathcal{M}_n = \{(x_1, y_1), \ldots, (x_n, y_n)\}$ (to which we will refer to as prior at step $n + 1$), and the subsequent measurement $(x_{n+1}, y_{n+1})$, the updated distribution (posterior at step $n + 1$) is

$$P_{n+1}(\lambda|\mathcal{M}_{n+1}) = \frac{P(y_{n+1}|\lambda, x_{n+1})P_n(\lambda|\mathcal{M}_n)}{P_n(y_{n+1}|x_{n+1})}. \tag{1}$$

The initial prior distribution $P_0(\lambda)$ is chosen arbitrarily, and it reflects our initial assumptions on the parameters of the system. Its choice balances the algorithm's behavior between improving prediction accuracy and reducing parameter uncertainty. With an unsuitable prior, many more measurements may be necessary to overcome the initial prejudice it conveys. A.3 shows some examples of the effects of this choice. Also, physical measurement errors may affect the algorithm's performance, if not handled properly. We discuss the effect of physical non-idealities in Appendix B.4

To choose the next measurement to perform, it is possible to define a query function, which assigns to each possible experiment $x$ its expected value: the $x$ for which the query function is maximized is the one that is expected to be the most useful to measure [47]. A possible choice of the query function is the expected information gain when measuring at $x$:

$$I_n[\lambda, y](x) = \int dy d\lambda P_n(\lambda) P(y|\lambda, x) \log \frac{P_{n+1}(\lambda|y, x)}{P_n(\lambda)}. \tag{2}$$

This can be interpreted as the mutual information between $y$ and $\lambda$ given a measurement at $x$, or in other words, as the entropy reduction after measuring $(x, y)$.

However, Eq. 2 is hard to estimate. First, it requires to be able to sample from the prior distribution $P_0(\lambda)$, and second, it requires the value itself of the posterior probability distribution $P_1(\lambda|y_0, x_0)$, since it appears inside the logarithm. In principle, we could obtain the posterior distribution with the Bayes rule, Eq. 1, but it is not efficient to calculate its normalization factor $P_0(y|x) = \int d\lambda P_0(\lambda) \log P_0(\lambda)$. Overall, to get Eq. 2 we would require two nested Monte Carlo estimates, one for the evidence and one for the mutual information estimation. Then, we should optimize over $x$ to get the best measurement to perform.

Modern neural network techniques allow implementing variational bounds and perform a much more efficient estimate [28], amortizing the cost of the computation. For example, we can avoid evaluating

the posterior [17] explicitly by introducing a new function $Q(\lambda|y, x)$ and calculating the quantity

$$I(x) = \int dy d\lambda P_0(y|x) P_1(\lambda|y, x) \log \frac{Q(\lambda|y, x)}{P_0(\lambda)}. \tag{3}$$

If $Q$ is a probability distribution, we know that $KL(P_1(\lambda|y, x)||Q(\lambda|y, x)) \geq 0$, thus $I(x) \leq I_0[\lambda, y](x)$. This is the so-called Barber-Agakov bound of mutual information [3]. We can parametrize $Q(\lambda|y, x)$ with a neural network, so that we do not need to find a different $Q$ function for every different $(x, y)$, but we can interpolate, amortizing the costs of the evaluation.

In particular, we use a conditional normalizing flow [54]: we start from a normal Gaussian distribution $\mathcal{G}(z)$ and perform a series of invertible transformations (parametrized by neural networks) having a Jacobian that is simple to estimate. Optimizing over $Q(\lambda|y, x)$ means to optimize over the parameters of the neural networks that implement the transformations. Normalizing flows can be easily implemented in common deep learning frameworks like TensorFlow [37]. In particular, we use TensorFlow Probability [11]. Please refer to A for more details.

There are multiple ways to estimate the performance of the proposed result (and compare with alternative techniques). The sum of the information gained at step $n+1$ after measuring $(x_{n+1}, y_{n+1})$,

$$\int d\lambda P_n(\lambda|y_n, x_n) \log \frac{Q(\lambda|y_n, x_n)}{P_n(\lambda)}, \tag{4}$$

can be a measure for how much we understood about the system. However, in contrast to (3), which represents the expected information gain averaged over all the possible measurement outcomes, this quantity can also be negative. Indeed, it is possible that an unexpected outcome of a measurement increases the global uncertainty on the parameters. Properties of the final posterior distribution such as its variance, or the comparison of its mean (or most likely value) with the true parameters are also a possible metric to consider. On the other hand, the best prediction for an observation is actually the average over all the allowed parameters

$$P(y|x) = \int d\lambda P_n(\lambda|\mathcal{M}_n) P(y|\lambda, x). \tag{5}$$

It is possible to compare this prediction with the likelihood calculated at the true parameters $\lambda^*$, for example using the Kullback-Leibler divergence. This gives an idea of how much the predictions of our model can differ from the observations. It is important to emphasize that if the physical model is correct, we would expect only a single parameter set to be relevant, corresponding to the true parameters of the system. However, if the model does not represent well the true system, the use of multiple values may be helpful to produce a better approximation.

## 4  Applications and discussion

We have already mentioned in the introduction the wide variety of quantum technology platforms, system parameters, and potential measurement approaches that could benefit in principle from active learning approaches. In the following, we apply the technique to the characterization of two illustrative quantum platforms. In both cases, the behavior of the device can be described by some unknown parameters that we want to determine through experiments. While many experiments would be possible, for the sake of simplicity, we fix the measurement strategy so that different experiments only differ in some experimental parameter we can tune. The role of Deep Bayesian experimental design we want to show is two-fold: first, it provides a way to integrate the outcome of a new measurement into previous knowledge; second, it suggests the best measurement to perform next. Appendix B provides the technical details of our implementation and a simple illustrative application, while Appendix C shows a comparison with alternative active learning approaches.

### 4.1  Coupled cavities

As a first example, we consider a linear device, where wave transmission is measured to extract the resonance frequencies of a coupled-cavity array. Such cavity arrays [23, 25, 40, 1] have been considered in studies of transport as coupled-resonator optical waveguides in optical setups or quantum many-body physics of photons, when combined with nonlinear elements like qubits.

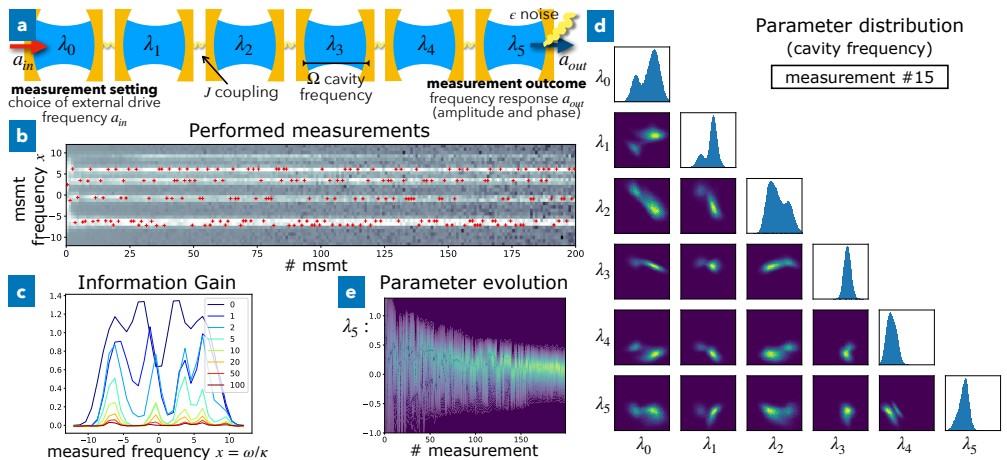

Figure 1: Example application to estimating the frequencies of an array of 6 coupled cavities. (a) Sketch of the system. (b) Measurements at each step. The red symbols show the chosen frequency $x$ at each step. In the background, the value of the expected information gain for each possible $x$ (brighter is higher), normalized at each step, i.e. $IG(x)/\max_x IG(x)$. (c) Expected information gain for each possible measurement $x$. A peak represents an optimal value. Different lines represent different measurement steps. (d) Inferred parameter distribution after 15 measurements. The diagonal shows the marginal distribution $P(\lambda_i)$, the off-diagonal plots the two-variable slices $P(\lambda_i, \lambda_j)$. (e) Evolution of the marginalized posterior of the last cavity frequency.

We consider a cavity array with the following Hamiltonian:

$$\hat{H} = \sum_j \Omega_j \hat{a}_j^\dagger \hat{a}_j + \sum_j J_{j+1}(\hat{a}_j^\dagger \hat{a}_{j+1} + \text{h.c.}), \tag{6}$$

where the sum runs over all but the last of the cavity modes in this chain with open boundary conditions. Here and in what follows, we consider $\hbar = 1$.

Since the setup we are dealing with is linear, it is sufficient to solve the classical equations of motion for the coupled modes, driven by a wave entering from a waveguide coupled to the first cavity. To this end, we consider the classical coherent-state amplitudes $a_j$ corresponding to the quantum operators $\hat{a}_j$, including drive and decay as prescribed by input-output theory [8]. For brevity of our notation, we collect all these amplitudes in a vector and all frequencies in a matrix, where first-neighbour interactions are $J$ and proper frequencies $\Omega_i$. Let our system also have some internal decay $\kappa_{\text{int}}$ and external decay $\kappa_{\text{ext}}$, which only applies to the cavities coupled to the environment, for example the first and last.

Given the entering fluctuating field $a_{\text{in}}$, which can also contain a laser drive, the output field $a_{\text{out}}$ is [9]

$$a_{\text{out}} = \mathcal{S} a_{\text{in}} = \left( \mathbf{1} - \frac{\sqrt{\kappa_{\text{ext}}}}{-i(\omega - \Omega) + \frac{\kappa}{2}} \right) a_{\text{in}}. \tag{7}$$

This equation represents the response of the system to an external perturbation. In particular, as shown in 1a, we choose a measurement strategy in which we drive the first cavity at a tunable frequency (i.e. we send a monochromatic wave) and measure the frequency response at the last cavity (i.e. the output wave at the same frequency). This gives us the scattering matrix element for transmission from the first to the last cavity, given by the $\mathcal{S}_{0N}$ in (7). Being it a complex number, its knowledge corresponds to measuring both the amplitude and phase of the emitted light. By performing the smallest possible number of measurements at a frequency $\omega$ close to the resonance frequency, we want to characterize the system by discovering the resonance frequencies of all the cavities. For example, they may all be close to a common resonance frequency, but slightly detuned (e.g. because of material fabrication imperfections). Each measurement is affected by noise $\epsilon$, which can be assumed to be Gaussian [8]. This noise, being observed in a measurement of the field amplitude, ultimately arises from quantum vacuum noise being injected into the input ports of the device (and through the dissipation channels) and propagating linearly through the setup.

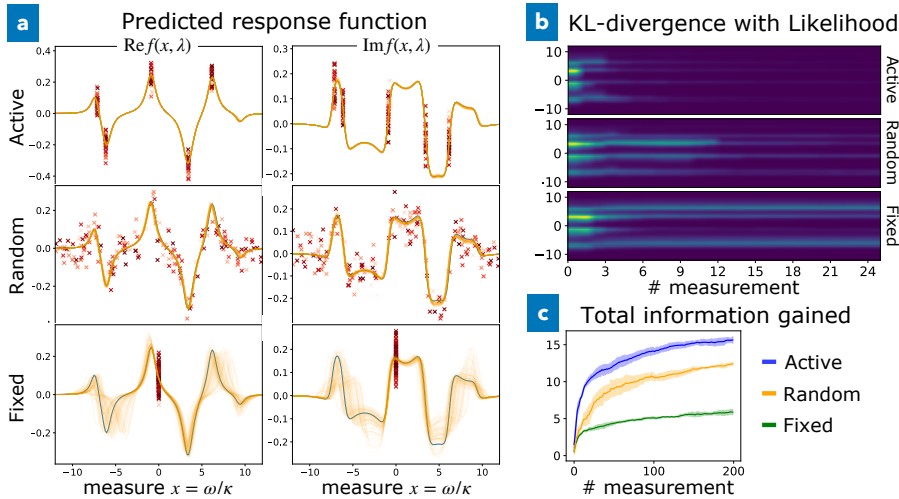

Figure 2: Comparison of different strategies for the choice of the next measurement setting for the coupled cavities system. (a) Final response after inferring the parameters with the active, random, or fixed strategy. The orange curves represent the response induced by various samples of parameters $\lambda$ from the final posterior. In blue, the true response. Red symbols mark the performed measurements and their outcome. (b) Evolution of the KL-divergence between the inferred response distribution $P(y|x)$ and the likelihood during the various measurements. Brighter means higher. (c) Cumulative sum of information gained after each measurement, i.e. $\sum_{n=1}^{N} \log P_n(\lambda|y, x) - \log P_{n-1}(\lambda)$. Higher values represent learning more about the system. Results are averaged over 3 runs.

From a classical perspective, we can imagine this system as a one-dimensional chain of springs (i.e. oscillators with some damping), each with its proper elastic constant and mass, which we want to determine. For this purpose, we make the first spring oscillate with a chosen frequency and observe how this perturbation sets the last spring in motion - in particular, how smaller its amplitude is, and what is the phase difference with respect to the drive.

We consider 6 coupled cavities, with each coupling uniformly chosen in $J_j/\kappa \in [1, 3]$. We assume a setting where the couplings $J$ and the dissipation $\kappa$ have already been calibrated and are therefore known, and our goal is to find the detunings. In the general notation of section 3, the measurement setting $x$ corresponds in this case to the choice of the drive frequency $\omega$, the response function $f(x, \lambda)$ corresponds to the scattering matrix element $\mathcal{S}_{0N}$, and the unknown parameters of the system $\lambda$ are the frequencies of the cavities $\Omega_i$.

In Figure 1, we show our results. As more measurements are performed, we improve our estimate of the cavity frequencies. By looking at the measurement frequencies that the active approach selects, in Figure 1b, we can identify a pattern, or measurement strategy: it measures the frequencies where the slope of the response is larger, alternating between different points. We see, in Figure 1c, that the information provided by new measurements decreases as we make more observations. At the same time, the posterior distribution converges to a sharp peak around the true values. For example, Figure 1e shows the evolution of the last cavity frequency distribution. It is important to emphasize that even if the posterior eventually converges to a Gaussian, it is not Gaussian at the beginning, as shown in Figure 1d. This is why it is necessary to approximate the posterior using a neural network approximation. These points give indeed the largest information on the intrinsic frequencies of the cavities.

We compare the results of the inference with two other strategies: fixed and random. The fixed strategy is the simplest: all the measurements are performed at the same $x$ value, and it is clearly often not possible to learn all the parameters from that. There will usually be a region of the response function, close to the measurement region, that is very well learned, while other regions will not be very accurate. A better naive measurement selection approach is the random one. In this case, measurements are chosen uniformly within the specified measurement range (here, $x = \omega/\kappa \in [-12, 12]$). We can see that in Figure 2: the active strategy, i.e. the one that chooses the next measurement greedily by maximizing the expected information gain, learns the parameters of the system faster than the random

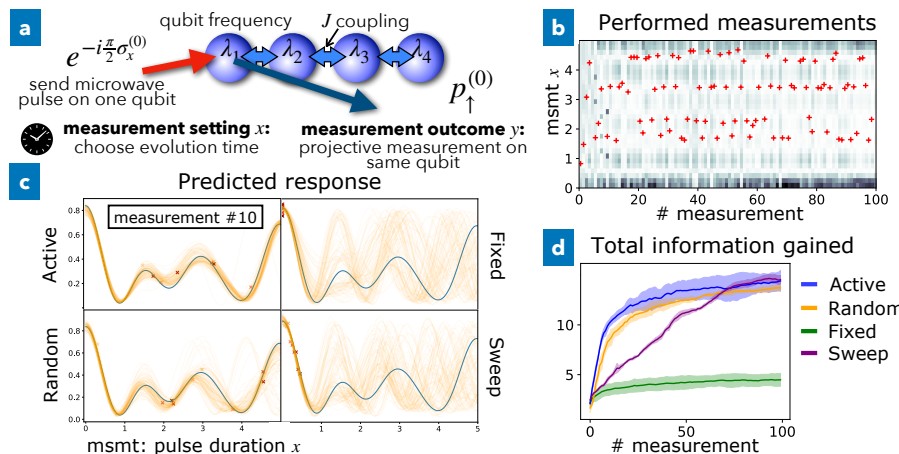

Figure 3: Example application to a 4-qubit array. (a) Sketch of the system. (b) Chosen pulse duration $x$ at each step. The red symbols show the chosen $x$ at each step. In the background, the expected information gain for each $x$ (brighter is higher), normalized at each step maximum. (c) Response after 10 measurements, comparing the active, random, fixed and uniform approaches. The orange curves represent the response induced by sampling from the final posterior. In blue, the true response. Red symbols mark the performed measurements and their outcome. (d) Sum of information gained after each measurement. Results are averaged over 3 runs.

one. Asymptotically, both the random and active strategy will converge to the right values. However, active selection allows to save a lot of measurements.

## 4.2 Array of qubits

In our second example, we turn to an array of qubits, which is a widespread platform used for quantum simulations and quantum computing [25, 4, 55, 36, 1, 56]. We consider an array of $N$ qubits and want to discover their frequencies $\omega_i$. The many-body Hamiltonian reads like

$$H = \sum_{i=0}^{N-1} \omega_i \sigma_z^{(i)} + J \sum_{i=0}^{N-2} \sigma_x^{(i)} \sigma_x^{(i+1)}. \tag{8}$$

As a measurement strategy, we consider a typical pulsed scheme followed by a projective measurement to extract qubit parameters. At the beginning, the system starts in its ground state $|\psi_0\rangle$. We assume that qubits can be individually addressed, and can be subjected to microwave pulses and projective measurements. In particular, we excite the first qubit of the array by applying a microwave pulse with pulse area $\pi$. This corresponds to implementing the unitary $U = \exp -i\frac{\pi}{2}\sigma_x^{(0)}$. We let the system evolve freely for some time, and then we perform a projective measurement on the first qubit. The evolution allows for the propagation of the external excitation. In each experiment, we can choose the duration of this free evolution after the initial pulse, which we will denote as the "measurement pulse duration" $x \in [0, 5]$. The outcome of the projective measurement is binary, according to probability

$$p_\uparrow^{(0)} = \langle \psi(t)|\hat{\mathcal{P}}_\uparrow^{(0)}|\psi(t)\rangle, \tag{9}$$

where $\hat{\mathcal{P}}_\uparrow^{(0)} = |\uparrow\rangle_0\langle\uparrow|$ is the projection operator on the "up"-state of the first qubit (qubit number 0). We considered a 4-qubit system and performed 100 measurements in series, choosing the pulse duration for each of them (see 3a). In this case, each of those measurements is in itself a multishot measurement, being repeated 100 times to give better statistics (otherwise the outcome would only be either 0 or 1).

The reader unfamiliar with quantum mechanics can imagine this example as a chain of interacting objects, each with an unknown parameter to identify. This time, we make a one-time perturbation to the first element (we "flip" it). This perturbation propagates along the chain according to a quantum law. Finally, we measure the state of the first element after a time interval we can choose. The

probability of measuring it "up" or "down" (which are the only two possible outcomes) depends on how the initial perturbation gets propagated, thus on the internal parameters we want to learn.

In Figure 3, we compare the active strategy with a random and a fixed one, and show the efficiency advantages of the active one in terms of gained information. We also show the performance of a uniform strategy, which takes sequential equally-spaced measurements, i.e. sweeping the pulse duration from 0 to the maximum allowed value. It is interesting to observe ( 3b) that this strategy eventually learns the parameters of the system, but it requires many more measurements than the random one. Also, in Figure 3d we see that the active strategy usually requires, on average, sensibly fewer measurements than the random strategy to provide the same total information gained. For example, to reach a total information gained of $10$, we require about $10$ measurements with the active approach or about $20$ measurements with the random one. The fixed strategy always measures at $x = 0$, i.e. does not allow any relaxation after the microwave pulse on the first qubit. As a consequence, this measurement effectively only provides information about the interacting ground state of the system, which already contains some limited information about the qubit frequencies. We emphasize that since the interacting quantum many-body ground state is not an eigenstate of $\sigma_z$ acting on the first qubit, the probability of measuring $\uparrow$ at $t = 0$ is not $1$. Furthermore, it is possible to spot a sort of strategy in the pattern of the measurements performed by the active selection: a few measurement regions are alternated cyclically. Indeed, we can imagine that some regions give the largest information about some parameters. After we measure in one region, we increase the accuracy on some parameters, and another region becomes more relevant, until it is useful to measure again at the initial spot.

## 5 Outlook

In this paper, we have introduced deep optimal Bayesian experimental design for modern quantum technologies. This approach approximates the posterior update with a variational bound and a deep neural network and allows extracting the optimal measurement to perform at each step. We have shown the application of this technique to two promising quantum platforms, cavity arrays and qubit chains. In both cases, the active measurement selection technique allows learning the parameters of the system with fewer measurements than other strategies. The main challenge at present is the time it takes to update the neural network representing the Bayes posterior distribution as well as optimizing over possible measurement settings. This time is still too large in order for this technique to be deployed economically "as is" in the given scenarios, where individual measurements can happen on microsecond time scales, and even extended sequences used to reduce shot noise will probably not last longer than a millisecond for one measurement setting, except when extreme accuracy is called for. A more detailed analysis of computational costs is available in the Appendix D. In the present (not yet very optimized) technique, an optimization run still takes on the order of hundreds of seconds per measurement step, which is currently a drawback of Bayesian Optimal Experimental Design based on deep neural networks in general.

However, a remedy would consist in performing the optimization on a number of different example scenarios, in simulations, and then train a neural network in a supervised fashion to learn the suitable choice of the next measurement setting based on all the previous settings and outcomes. In this way, we could really develop a sort of policy that can be deployed to characterize unknown systems of the same class. A specific cost function may also be added to take into account that some measurement values may require more resources and thus be more expensive than others [47]. A further generalization can be to drop the likelihood assumption and try to develop a likelihood-free active inference technique. This would be similar to [7]. However, we do not want to discretize the input space $x$, since it would not scale up to large dimensions. Finally, we notice that we are employing a greedy strategy which always chooses the single next best measurement to perform. Alternative approaches may include providing an overall measurement budget (e.g. total number of measurements) or set a target total information gain to achieve. It would be interesting to study how these sequential strategies [15], perhaps approximated with a reinforcement learning agent [48, 14], may suggest better measurement strategies than the greedy one we applied in this paper.

From the broader perspective, we can imagine in the future to generalize the measurement setting $x$ to become an entire experimental setup, and to perform the search on a broader class of experiments. This would be a very useful aspect of a future "artificial scientist" that tries to explore the world and learn a model by performing experiments autonomously.

The code of this paper and data related to the experiments is available on GitHub [2] and Zenodo[3].

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

# A  Bayesian Experimental Design

In this Appendix, we describe in more detail the standard framework of Bayesian experimental design.

## A.1  The Bayesian parameter update

Starting from some a priori distribution $P_0(\lambda)$, that reflects all our expectations on the parameters of the system (e.g. parameter range, symmetries, constraints), we can update it according to the Bayes rule [42]

$$P_1(\lambda|x_0, y_0) = \frac{P(y_0|\lambda, x_0)P_0(\lambda)}{P_0(y_0|x_0)},\tag{10}$$

where $(x_0, y_0)$ is the first measurement and its outcome and

$$P_0(y_0|x_0) = \int d\lambda P(y|\lambda, x)P_0(\lambda).\tag{11}$$

The distribution $P_1(\lambda)$, which depends on the performed experiment and on its outcome $(x_0, y_0)$, is called posterior distribution. It represents the updated knowledge on $\lambda$ after taking into account the result of the experiment. It is easy to show that in case of multiple (independent) experiments, $\{x_i, y_i\}^n$, this equation can be easily generalized to a recursive form:

$$P_n\left(\lambda|\{x_i, y_i\}^n\right) = \frac{\prod^n P(y_i|\lambda, x_i)P_0(\lambda)}{\int d\lambda' \prod^n P(y_i|\lambda', x_i)P_0(\lambda')} = \frac{P(y_n|\lambda, x_n)P_{n-1}\left(\lambda|\{x_i, y_i\}^{n-1}\right)}{P_{n-1}(y_n|x_n)}\tag{12}$$

with

$$P_n(y_n|x_n) = \int d\lambda P(y_n|\lambda, x_n)P_{n-1}\left(\lambda|\{x_i, y_i\}^{n-1}\right).\tag{13}$$

As one would expect, we get exactly the same updated prior if we perform many experiments in parallel and then update our knowledge or if we perform them one after the other and update our prior after each single experiment.

## A.2  The information gain query function

We define a query function to choose which experiment to perform. This function assigns to each possible experiment $x$ its expected usefulness: the $x$ for which the query function is maximized is the one expected to be most useful to measure [47]. The query function we use in this paper is the expected Kullback-Leibler divergence between the updated parameter distribution and the prior. We start from

$$KL(P_1(\lambda|y, x)||P_0(\lambda)) = \int d\lambda P_1(\lambda|y, x) \log \frac{P_1(\lambda|y, x)}{P_0(\lambda)},\tag{14}$$

which estimates how much the prior differs from the updated distribution as a function of the experiment $x$ and its outcome $y$. Then, we calculate its average over the possible $y$, which gives the expected information gain when measuring at $x$ i

$$IG(x) = \int dy P_0(y|x)KL(P_1(\lambda|y, x)||P_0(\lambda)) = \int dy d\lambda P_0(y|x)P_1(\lambda|y, x) \log \frac{P_1(\lambda|y, x)}{P_0(\lambda)}.\tag{15}$$

This equation can be rewritten as

$$\int dy d\lambda P_0(\lambda)P(y|\lambda, x) \log \frac{P_1(\lambda|y, x)}{P_0(\lambda)} = H_0(\lambda) - H_1(\lambda|y)(x) = I_0(\lambda, y)(x),\tag{16}$$

with $H_0(\lambda) = -\int d\lambda P_0(\lambda) \log P_0(\lambda)$ and $H_1(\lambda|y)(x) = -\int P_0(\lambda)P(y|\lambda, x) \log P_1(\lambda|y, x)$. This can be interpreted as the mutual information between $y$ and $\lambda$, or, in other words, as the entropy reduction after measuring $(x, y)$. The optimal $x$ to measure next, if we consider a greedy strategy, is the one that maximizes this quantity.

Figure 4 shows a summary of the described procedure.

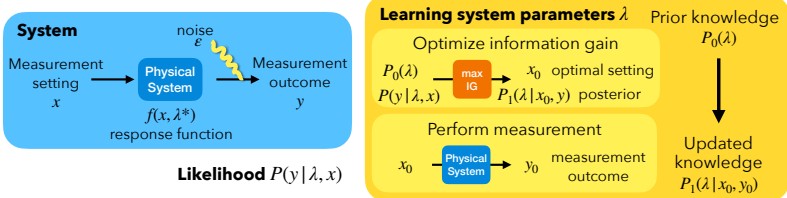

Figure 4: Sketch of the model of a system's response and summary of the parameter estimation procedure. On the left: the observed system produces a noisy observation given the measurement settings and the system's parameter $\lambda$. On the right: starting from our prior, we perform multiple steps. In each step, given the last prior and the likelihood, we approximate the posterior by minimizing the Barber-Agakov bound Eq. 3. This minimization also gives us the optimal measurement setting to choose. After performing the measurement, the approximated posterior, conditioned on the measured value and outcome, becomes the new prior for the next iteration step.

### A.3 Choice of the prior

To give an idea of the effects of the choice of the prior, we consider again the example of the coupled-cavity arrays in the main text. There, the prior is an independent Gaussian distribution with zero mean and unit variance, and this is the same distribution from which we sampled the true parameters of the system.

Figure 5 compares this choice with two different choices: a biased prior that is focused on the wrong mean, and a broader one which has a very large variance. We see that the posterior starts approaching the correct values, but it requires more measurements. In the case of the biased prior, we see that the active strategy will exploit the surprise of observed outcomes to improve accordingly, while the random strategy hardly learns a good response with the same number of measurements. On the other hand, it is much easier to reduce the variance of a broader prior to converge towards the true parameters.

If the procedure is interrupted too early, before the desired accuracy is obtained, the posterior mean might suggest different parameters from the true ones, or have multiple modes around different candidates. This is because those candidates might produce a response function similar enough to the system's response; with additional measurements, the posterior shifts towards the true parameters, unless the other ones really represent a symmetry of the system, or the measurement procedure does not allow to distinguish a system with the proposed parameters and the true ones.

From the numerical perspective, it is important that the prior keeps a sensible overlap with the approximate posterior distribution $Q(\lambda|y,x)$. Indeed, to estimate the information gain with the Barber-Agakov bound in (2), the product $P_0(\lambda)\log Q(\lambda|y,x)$ matters. If the approximate posterior $Q(\lambda|y,x)$ is too small for some values of the prior, divergences can arise, while if the prior is too small in the regions where the approximate posterior has most of its density, they might not be sampled with the Monte Carlo estimate and no dependence between the estimated information gain and measurement setting $x$ will be detected. A broader prior can help to solve the former problem, at the expense of requiring more samples in the Monte Carlo estimate to prevent the latter. For example, in our experiments with the coupled cavities, a prior with $\mu = 10$ and $\sigma = 0.05$, or a prior with $\mu = 0$ and $\sigma = 100$ almost always breaks down the numerical precision of our implementation.

## B Applications

### B.1 Implementation

In our experiments, we employ a 4-layer normalizing flow, each employing a 2-layer MADE network (Masked Autoregressive flow for Density Estimation [41]) with 64 neurons each, for a total 20544 parameters. The MADE network is conditioned on the $y$ value, allowing to amortize the posterior approximation for all possible measurement outcomes. On the other hand, the posterior model does not explicitly depend on the measurement choice $x$: we obtain the posterior associated to the optimal measurement $x^*$ by optimizing the Barber-Agakov bound.

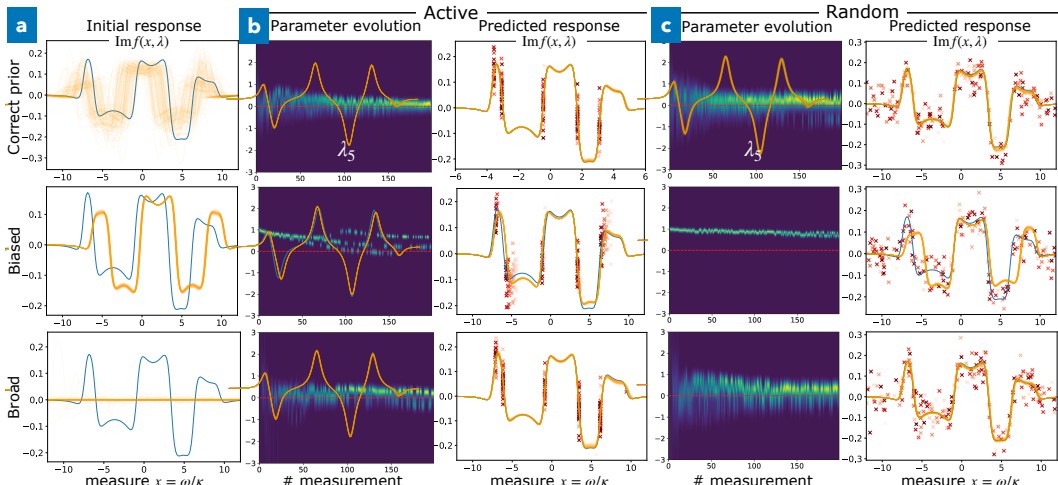

Figure 5: We compare the learning of the parameters of the coupled-cavity example in the main text (Gaussian prior with $\mu = 0$ and $\sigma = 1$) with the case of a biased prior ($\mu = 1$ and $\sigma = 0.05$) and that of a much broader prior ($\mu = 0$ and $\sigma = 10$). (a) Initial response $P_0(y|x)$ using the chosen prior distribution. For compactness, we only show the imaginary part of the response. (b) Learning with an active strategy. Left: Evolution of a parameter distribution, $\lambda_5$, as an example. The red line highlights the true value. Right: Predicted response function after 200 measurements. (c) Learning with a random strategy. Same plots as in (b). As the parameter distribution evolves, the predicted response function improves. However, it takes many more measurements to reject values that produce similar responses but are not the true ones. In the case of the biased prior, the choice of alternative values of $\lambda_5$ correlate with alternative choices of $\lambda_0, \ldots \lambda_4$ to compensate for the wrong value. To draw overall conclusions on the learning, one should analyze the full posterior. The strong initial bias does not vanish completely even after 200 measurements. Especially with the random approach, the predicted response is still very different from the truth. By looking at the behavior of the active strategy, we see a shift in the chosen measurement settings that reflect the updated parameter knowledge in the case of the biased prior; in the case of the broader one, it initially starts with some random measurements until it identifies the regions of the response with the largest slope and starts exploiting them.

Starting from a random input $z$ sampled from a normal Gaussian, each MADE layer transforms the distribution until approximating the posterior $Q(\lambda|y, x)$. We can imagine a MADE layer as a fully-connected neural network $f_\theta(z, y)$ taking the previous random input $z^{(i)}$ at layer $i$ of the normalizing flow and the condition on the measurement outcome $y$ and producing outputs $\mu(\lambda|y)$ and $\sigma(\lambda|y)$. The final output is $z^{(i+1)} = f_\theta(z^{(i)}) = \mu(z^{(i)}, y) + e^{\sigma(z^i|y)} z^{(i)}$. We employ ReLU activation functions for all the 4 layers of the MADE but the last one.

The MADE network is a dense network with a masked structure, so that the $n$th outputs $\mu_n$ and $\sigma_n$ only depend on the previous $1 \ldots n - 1$ inputs. In this way, the Jacobian determinant needed to calculate the probability density from the initial Gaussian is efficient to evaluate (being it a triangular matrix) [41]. Therefore, we can easily sample and evaluate probability densities of the approximate posterior. With respect to the original implementation [41], we activate the $\sigma$ with $a \tanh(\sigma/a)$, with $a = 5$ to prevent numerical explosion or collapse of the distribution variance and increase stability.

After each layer of the MADE network, the input vector is shifted by one with a permutation layer (i.e. $x_1 \ldots x_n \to x_n, x_1, \ldots x_{n-1}$), so that the subsequent layer applies a masked transformation to a different sequence of the input. Figure 6 shows a sketch of the architecture. We optimize the network with an Adam optimizer with learning rate $\eta = 10^{-3}$.

In terms of memory, the network we used has roughly 20000 parameters, so it is relatively small compared to available memory on current GPUs. Especially in the case of the qubit system, the main effort lies in the simulation, and the memory consumption grows with larger qubit systems.

The system simulation runs on the GPU to exploit its parallelization capabilities and to allow differentiating through the response function. Also, we repeat the optimization over $x$ in parallel on

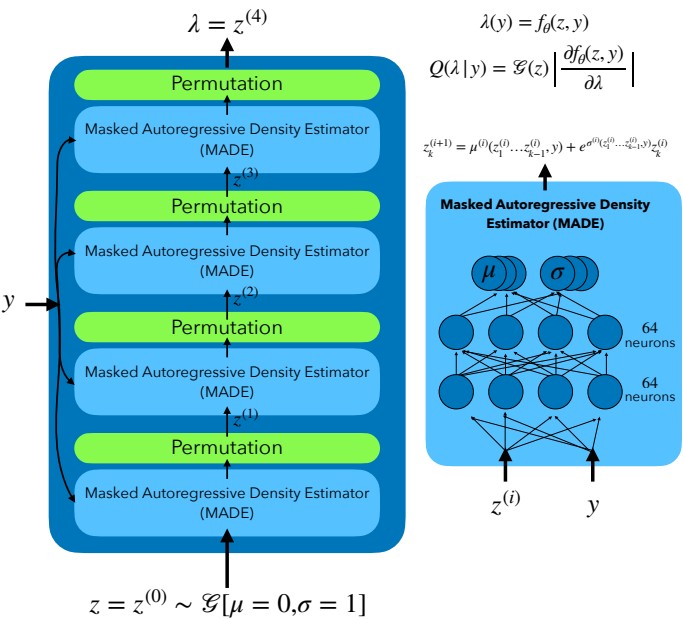

$$\lambda = z^{(4)}$$

$$\lambda(y) = f_\theta(z, y)$$

$$Q(\lambda \mid y) = \mathscr{G}(z) \left| \frac{\partial f_\theta(z, y)}{\partial \lambda} \right|$$

$$z_k^{(i+1)} = \mu^{(i)}(z_1^{(i)} \dots z_{k-1}^{(i)}, y) + e^{\sigma^{(i)}(z_1^{(i)} \dots z_{k-1}^{(i)}, y)} z_k^{(i)}$$

$$z = z^{(0)} \sim \mathscr{G}[\mu = 0, \sigma = 1]$$

Figure 6: Sketch of the normalizing flow used to approximate the posterior distribution $Q(\lambda|y, x)$. Left: normalizing flow architecture $f_\theta(z, y)$, which transforms a random normal input $z$ to a sample from the posterior, $\lambda$, conditioned on the observed outcome $y$ (see the main text). Right: structure of a MADE layer. It takes as input the output of the previous layer and the observed outcome $y$ and outputs $\mu$ and $\sigma$, which can be used to perform an invertible transformation to the input. In particular, we remind that for the transformation to be invertible, we need to constrain the network connectivity so that each component of the output only depends on the previous components. As a consequence, since the first component can only be rescaled independently of all the other components, we need a permutation layer after each MADE layer.

multiple $GPU$s, starting from different initial values, and select the final $x$ with the largest information gain. This allows avoiding getting stuck in local minima and select suboptimal measurement settings. Alternative and smarter machine learning regularization techniques could be used instead for a more resource-efficient approach.

## B.2 An illustrative example

To give an idea of how the presented technique works, we analyze a case where many calculations can be done analytically in closed form [45]. We consider a linear system with parameters $\lambda = (\lambda_1, \lambda_2)$ and response function

$$f(x) = \hat{x} \cdot \lambda = \lambda_1 \cos\theta + \lambda_2 \sin\theta \tag{17}$$

with $x = (\cos\theta, \sin\theta)$. By measuring at a chosen angle $\theta$ we observe the projection of $\lambda$ in the given direction and want to reconstruct the $\lambda$ vector. Each observation is corrupted by some Gaussian noise $\epsilon$ with zero mean and $\sigma_\epsilon^2$ variance, so that

$$y = f(x) + \epsilon. \tag{18}$$

This implies that the likelihood of the system is also Gaussian:

$$P(y|x, \lambda) = \mathcal{G}(f(x), \sigma_\epsilon). \tag{19}$$

As a consequence, starting from a Gaussian prior, also the posterior will be Gaussian, and it can be expressed analytically: given the prior

$$P_0(\lambda) = \mathcal{G}(\mu_0, \Sigma_0), \tag{20}$$

it is easy to show that the posterior reads

$$P_1(\lambda|x_0, y_0) = \mathcal{G}(\mu_1, \Sigma_1), \tag{21}$$

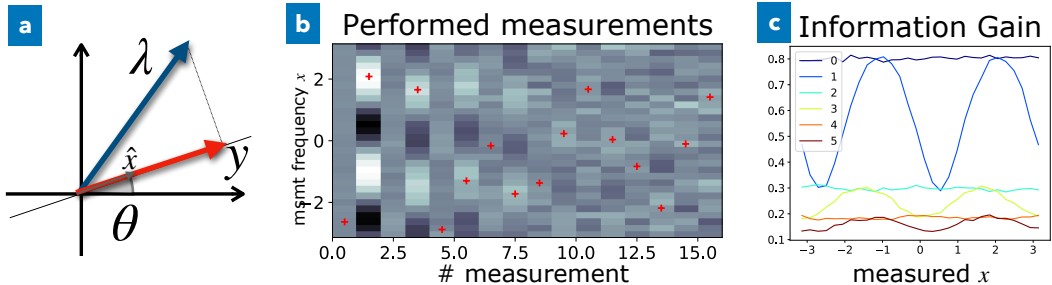

Figure 7: Simple toy model. (a) Sketch of the system: at each step, we choose an angle $\theta$, which defines the unit vector $\hat{x} = (\cos\theta, \sin\theta)$, and observe $y = \lambda \cdot \hat{x} + \epsilon$. (b) Measurements chosen with an active measurement strategy (red dots). In the background: information gain as a function of $x$. (c) Expected information gain curves $IG(x)$ for the first 5 measurements. We see that even measurements can usually be chosen arbitrarily, but the odd measurements are optimal when shifted by $\pi/2$ from the previous.

with

$$\mu_1 = \Sigma_1 \left( \frac{x_0 y_0}{\sigma_\epsilon^2} + \mu_0 \Sigma_0^{-1} \right), \tag{22}$$

$$\Sigma_1^{-1} = \frac{\chi(x_0)}{\sigma_\epsilon^2} + \Sigma_0^{-1}, \tag{23}$$

and

$$\chi(x_0) = \begin{pmatrix} \cos^2\theta_0 & \cos\theta_0 \sin\theta_0 \\ \cos\theta_0 \sin\theta_0 & \sin^2\theta_0 \end{pmatrix}. \tag{24}$$

Iteratively, the posterior can become the new prior and the equation can be applied for all the subsequent measurements. Essentially, at each step the mean and the covariance matrix of the Gaussian prior get updated. We can use this system to compare the predictions of the neural network estimator with the explicitly calculated values.

Figure 7 shows a sketch of this simple system and gives a hint on the optimal strategy. Since at each measurement we can choose the direction on which we project $\lambda$, it is important to alternate different directions to learn both components. We see that there is no particular preference for the first measurement. But after the first measurement, it is useful to measure the orthogonal direction. Subsequently, it becomes degenerate again, and it keeps alternating random measurements with measurements shifted by 90 degrees.

We compare the active strategy with a random selection and a fixed one in Figure 8. The fixed strategy is always performing the same measurement $\theta = 0$. We see that in this way we can learn one component of the parameters, i.e. $\lambda_1$, but not the other. Indeed, the variance of $\lambda_2$ does not decrease when we perform new measurements as they give no information about it. As a consequence, we also see that the response function we learn is very uncertain outside the region that we always observed. Also, the total information gained with this strategy is quite smaller than the other ones. Regarding the active strategy, we can recognize a pattern. We see that it alternates random measurements with measurements shifted by 90 degrees from the previous one. Intuitively, as one measurement decreases the uncertainty in one direction, it becomes more useful to also decrease it in the orthogonal direction than to measure again at the same spot. However, we see that in this case the overall performance is not very different from always choosing the measurements randomly.

Finally, this example is also useful to test the neural network approximation of the posterior. Indeed, as previously stated, in this case we can easily perform the Bayesian update numerically and compare the results. Figure 9 shows a good compatibility between the active learning with the variational bound and the numerical calculation. We notice that, since the posterior is much more flexible than the Gaussian distribution we use for the numerical calculation, the mean values tend to fluctuate more with the neural network approach. This is because the Gaussian assumption is inherently correct in this simple example and the variational approach needs some steps to converge to it.

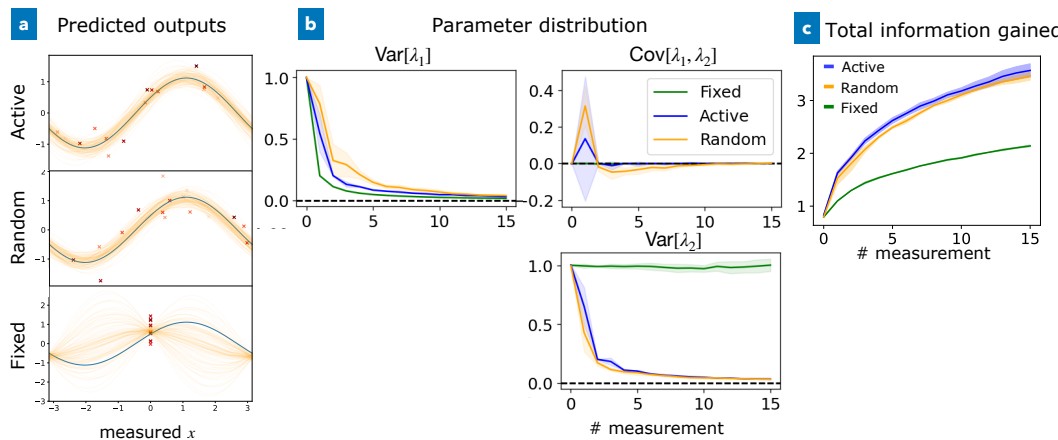

Figure 8: Comparison of active, random and fixed measurement choice for a simple toy model. (a) Comparison of the predicted response function after 15 measurements. The blue curve is the true response, the orange curves are sampled from the final parameter distribution. Red crosses indicate measurement outcomes. (b) Covariance matrix of the posterior distribution at each step. We see that the uncertainty on the parameters decreases as we learn more about the system. (c) Comparison of total information gained while observing the system using different approaches. The active strategy does not show a much stronger performance than random in this case.

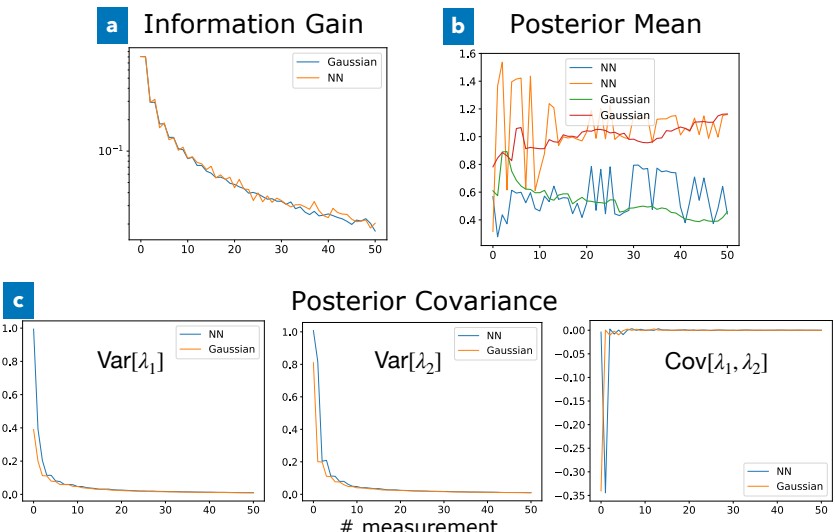

Figure 9: Testing the accuracy of the neural network posterior approximation against the numerical calculation for the Gaussian toy model. Both techniques perform updates at each step using the same sequence of measurements and outcomes for consistency. (a) Comparison of the expected information gain at each step. (b) Mean value of the parameter posterior at each step. Respectively, blue and orange for the first and second component of the neural network prediction, green and red for the numerical calculation. (c) Variance and covariance of the parameter posterior at each step.

### B.3 Coupled-cavity arrays

We consider a cavity array with the following Hamiltonian:

$$\hat{H} = \sum_j \Omega_j \hat{a}_j^\dagger \hat{a}_j + \sum_j J_{j+1}(\hat{a}_j^\dagger \hat{a}_{j+1} + \text{h.c.}), \tag{25}$$

where the sum runs over all but the last of the cavity modes in this chain with open boundary conditions. We consider the classical coherent-state amplitudes $a_j$ corresponding to the quantum operators $\hat{a}_j$, including drive and decay as prescribed by input-output theory [2]. Given first-neighbour interactions $J_i$ and proper frequencies $\Omega_i$, we can define the matrix

$$\Omega = \begin{pmatrix} \Omega_0 & J_1 & 0 & \dots & \\ J_1 & \Omega_1 & J_2 & 0 & \dots \\ 0 & J_2 & \Omega_2 & J_3 & \dots \\ \vdots & & & \ddots & \\ & \dots & 0 & J_N & \Omega_N \end{pmatrix}. \tag{26}$$

We also include some internal decay $\kappa_{\text{int}}$ and external decay $\kappa_{\text{ext}}$, which only applies to the cavities coupled to the environment, for example the first and last. The overall decay can be represented by a vector

$$\kappa = \begin{pmatrix} \kappa_{\text{int}} + \kappa_{\text{ext}} \\ \kappa_{\text{int}} \\ \vdots \\ \kappa_{\text{int}} \\ \kappa_{\text{int}} + \kappa_{\text{ext}} \end{pmatrix}. \tag{27}$$

Let $a_{\text{in}}$ be the entering fluctuating field, which can also contain a laser drive, and $a_{\text{out}}$ the output field. The behavior of the system is described by the input-output relations [9]

$$\begin{cases} \dot{a} & = \left(-i\Omega - \frac{\kappa}{2}\right)a + \sqrt{\kappa_{\text{ext}}}a_{\text{in}} \\ a_{\text{out}} & = a_{\text{in}} - \sqrt{\kappa_{\text{ext}}}a \end{cases} \tag{28}$$

From the first equation, we can write the Green function (i.e. consider a drive $a_{\text{in}} \propto e^{-i\omega t}$ and assume also $a \propto e^{-i\omega t}$)

$$a = \frac{\sqrt{\kappa_{\text{ext}}}}{-i(\omega - \Omega) + \frac{\kappa}{2}}a_{\text{in}}. \tag{29}$$

Therefore, the final input-output relation becomes

$$a_{\text{out}} = \left(1 - \frac{\sqrt{\kappa_{\text{ext}}}}{-i(\omega - \Omega) + \frac{\kappa}{2}}\right)a_{\text{in}}. \tag{30}$$

For the case of spatially constant cavity mode frequencies, the $N$ eigenfrequencies of the open-boundary array span a bandwidth $4J$. All the frequencies in the bulk of the spectrum are doubly degenerate, and the spacing becomes smaller near the boundaries of the spectrum. This picture changes, and the degeneracies are broken, when we introduce variation in the onsite frequencies $\Omega_j$, which will be the generic situation we want to explore.

The system we discuss in the main text has couplings $J$ uniformly sampled in $[1,3]$ for each cavity, and proper frequencies sampled from a Gaussian distribution with zero mean and unit variance. The values for the specific example in Figure 1 and Figure 2 are shown in table 1. We optimize the posterior neural network for $8000$ steps after each measurement, using a batch size of $1500$ samples in (3).

### B.4 Effects of physical non-idealities

We briefly discuss the effect of physical non-idealities on the performance of the active learning approach. There are three aspects we can think of in this regard: (i) decay and decoherence, (ii) measurement errors, and (iii) model deviations.

Decay and decoherence in general are part of the model and therefore will be taken into account automatically in the active learning procedure discussed here, since they affect the likelihood.

| Parameters | Value |  | Known par. | Value |
| --- | --- | --- | --- | --- |
| (freq.) $\Omega_0$ | 1.040 |  | (coupling) $J_1$ | 2.733 |
| $\Omega_1$ | 0.326 |  | $J_2$ | 2.615 |
| $\Omega_2$ | 0.520 |  | $J_3$ | 1.956 |
| $\Omega_3$ | 0.900 |  | $J_4$ | 1.568 |
| $\Omega_4$ | −0.466 |  | $J_5$ | 2.620 |
| $\Omega_5$ | 0.004 |  | (int. diss.) $\kappa_{int}$ | 0.5 |
|  |  |  | (ext. diss.) $\kappa_{ext}$ | 0.5 |
|  |  |  | (msmt noise) $\epsilon$ | 0.05 |

Table 1: Parameters for the coupled cavity example discussed in the main text. On the left, the true frequencies we want to discover; on the right, the known parameters of the system.

Depending on the situation, decay rates may even be part of the parameters to be discovered or they can be assumed as given. Typically, the presence of decay and decoherence diminishes the variation of the measurement signal as a function of the underlying system parameters. This then has the tendency to decrease the expected information gain per measurement. One of our examples, the cavity array, illustrates that situation, where the slope of the response function decreases for stronger decay rates, eventually requiring a larger number of measurements. In particular, in Figure 10, we show the impact of different values of decay $\kappa$ on the performance of the active learning approach. In addition to the system analyzed in the main text, with $\kappa = 0.5$, we apply our technique also to a system with $\kappa = 1$ and $\kappa = 2$. In general, we see that systems with larger decay have lower information gain per measurement and more steps are required to learn the parameters. Also in these cases, even if the information gain is lower, an active approach can find measurement settings that still convey more information, and learn the parameters of the system more efficiently than with a random choice. We notice that in the case of largest decay, $\kappa = 2$, the active strategy may require at the beginning many more measurements to identify the measurements settings with the largest response slope (which provide the most useful measurements). By contrast, in the qubit example, we chose to assume no decay, since in realistic qubit systems decay times are much longer than the few oscillations considered here. If decay were included, this would produce a bias towards performing measurements at shorter times, where the dependence on the underlying system parameters is still maintained while such dependence will decrease to zero at long times due to the decay.

The measurement error is normally also included in the complete model and will affect the information gain per measurement. This is illustrated in the qubit example, where individual measurements have binary outcomes but $N$ repeated measurements then lead to a signal that carries a $1/\sqrt{N}$ shot noise contribution.

The most challenging type of non-ideality or error to be considered for any parameter discovery approach (whether active or not) are deviations between the assumed model and the ground truth model. In this case, no general statements can be made, but the Bayesian approach will simply fit the assumed model as well as it can to the observed data. In principle, one can remedy this by introducing new terms, weighted by additional parameters, into the model. If the prior distribution in these parameters is narrowly centered around zero, this will bias the procedure towards the simpler model while retaining the flexibility to discover a more complex model should the observations require this.

### B.5  Qubit chain

Table 2 shows the parameters employed for the qubit chain example. We optimize the posterior neural network for at least 2500 steps after each measurement, using a batch size of 700 samples in (3). When also optimizing over $x$ (i.e. optimal measurement setting), we keep optimizing the loss until the change in 500 steps is smaller than 0.05. This choice allows decreasing the training time for this more computationally-demanding system, since we train for a longer time only when required.

It is also interesting to look at how the prediction of the response function of a qubit system as in Figure 3 improves after the measurements suggested by an active strategy. As shown in Figure 11,

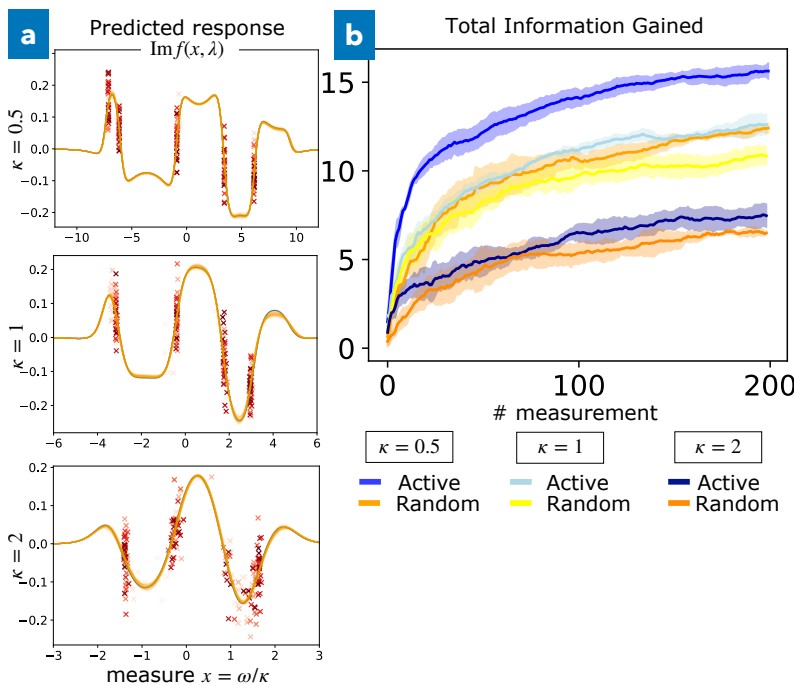

Figure 10: Effect of different decay values in the coupled-cavity example. The first plot, with $\kappa = 0.5$, corresponds to the system described in the main text. (a) Predicted response after 200 measurements using the active strategy. We only show the imaginary part. (b) Comparison of information gained at each measurement with both the active and random approach. Larger decay implies lower information gained per measurement and more measurements are needed to learn the parameters. Results are averaged over 3 runs.

| Parameters | Value |
|---|---|
| (freq.) $\omega_0$ | 1.163 |
| $\omega_1$ | 1.003 |
| $\omega_2$ | 1.045 |
| $\omega_3$ | 0.910 |

| Known par. | Value |
|---|---|
| (coupling) $J$ | 1.7 |
| (multishot) $n_{\mathrm{msmt}}$ | 100 |

Table 2: Parameters for the qubit chain example discussed in the main text. On the left, the true frequencies we want to discover; on the right, the known parameters of the system.

starting from a very uncertain prediction, the distribution of possible response functions converges to the true curve.

## C  Comparing with alternative active learning approaches

In this section, we show a comparison with other classical active learning approaches for regression.

The simplest approach to infer the parameters of a system is to apply the maximum likelihood principle [21]. By assuming a Gaussian error, we get the optimal parameters

$$\lambda^* = \arg\min_{\lambda} \sum_i (y_i - f(x_i, \lambda))^2. \tag{31}$$

We now need a new query function to select the next measurements to perform. Indeed, without a Bayesian approach, we don't have access to the posterior and thus to the information gain. One possibility is to assume a Gaussian uncertainty on the extracted parameters $\lambda^*$ and propagate this uncertainty to $P(y|x) = \int d\lambda P(y|\lambda, x) P_\lambda(\lambda)$. We can choose the next measurement $x$ as the one

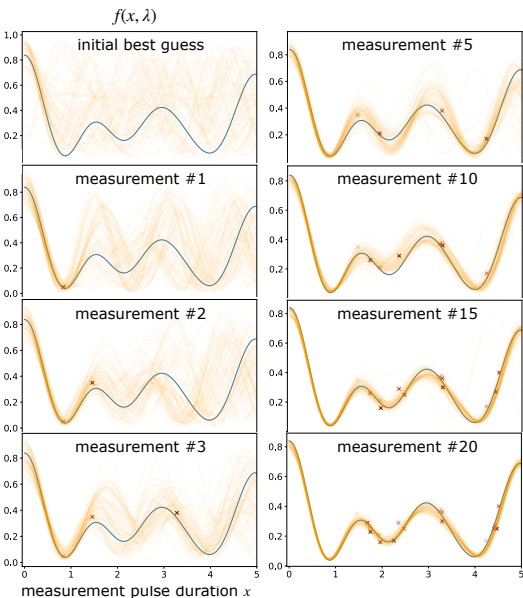

Figure 11: Prediction of the response function after a different number of steps. In orange, curves given by sampling parameters from the posterior; in blue, the true response function. The initial best guess is given by sampling from the prior distribution, before any measurement is performed.

that maximizes $\text{Var}[y|x]$, or use this quantity as weights for random sampling (e.g. with a Boltzmann distribution with tunable temperature).

Alternatively, we can consider $n$ different mean square optimization sets, $\lambda^{(k)}$, each obtained by performing the optimization on a different (randomly sampled) subset of the observations $\mathcal{M} = x_i, y_i$. Each $\lambda^{(k)}$ provides a response function $f(x, \lambda^{(k)})$, and we can consider the variance across the $n$ response functions as the query function to propose the next measurement.

If we wanted to ignore the known physics of the system, and just learn to predict the response function $y = f(x)$, we could use a Gaussian Process [5]. This method assumes that the function we want to fit, $f(x)$ has a Gaussian probability distribution with mean $\mu(x)$ and covariance $\Sigma(x, x')$. Each time we observe a new measurement, we can apply the Bayes theorem and update the mean and the covariance in closed form. A common assumption for the covariance matrix is to exponentially decay with the squared distance between points. We can choose the next measurement as the one where we have the largest uncertainty. We consider this case as a comparison, but we emphasize that, since we treat the device as a black box, we cannot use this technique to characterize the parameters of the system.

Figure 12 shows the results of our comparison in the case of the system of coupled cavities. Here we compare the mean squared error on the predicted response function (compared with the true one) as a function of the number of measurements. The approaches based on maximum likelihood take more steps to converge to the true parameters, since they implicitly assume a Gaussian posterior. This assumption typically does not hold, especially during the first steps. Furthermore, the least square optimization can be very unstable especially at the beginning, when the number of observations is smaller than the number of parameters to fit. Also, if the system had some symmetries and multiple parameters were equally valid, it would be harder to notice. We also believe that applications to situations with higher-dimensional $x$ (in this example, it was always one-dimensional) might further increase the advantage of a deep Bayesian approach.

## D    Towards practical applications

Depending on the application, different assumptions should be made, leading to different tradeoffs between the approximation precision and the time efficiency. The approach we propose is especially

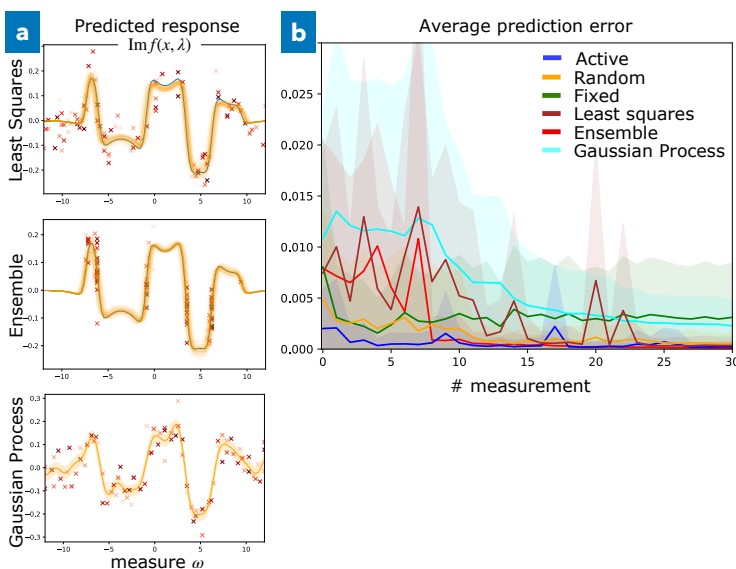

Figure 12: Comparison with various alternative active learning approaches. (a) Comparison of predicted response function and its uncertainty, (b) Mean squared error between the predicted response function and the true response (see text). Results are averaged over 3 runs.

advantageous in the setting where each measurement is very expensive and it is worth spending a long calculation time to really find the optimal next measurement setting.

The main computation time is split in two components: the estimation of the likelihood $P(y|\lambda, x)$ by simulating the system with various parameter settings, and the sampling of both the prior (which, after the first step, is also a normalizing flow corresponding to the posterior at the previous step) and the posterior. The simulation time mainly depends on the system. In the qubit array, for example, most of the effort is needed to diagonalize the Hamiltonian to exponentiate it for the time evolution. This time inevitably explodes with increasing number of qubits, as it happens with most quantum systems. The sampling time mainly depends on the size of the network: larger networks provide a more accurate approximation but are slower. In our examples, each training step took about $30ms$ in the case of the cavity example (for a batch size $1500$ samples; using $8000$ steps we need $240s$ per measurement) and $50ms$ (for a batch size of $700$ samples; using $2500$ steps we need $125s$ ) in the case of the qubits, on Quadro RTX 6000 GPUs. To give a feel for the time scaling, a system of $50$ cavities would take about $100$ ms per training step, while a qubit system with $6$ qubits needs more than $1s$ per training step.

For practical applications, some tuning will be helpful to increase the performance and reduce the time and memory requirements, according to the use case. For example, we empirically train the posterior network for a fixed number of steps. While this proved to be a good choice, smarter techniques such as choosing an adaptive number of steps and stopping when no change occurs for a given number of steps, or the use of an adaptive learning rate can allow for faster convergence. Also the batch size can be adjusted. To cut the training time, sometimes even a rough estimate of the information gain (with a smaller network or fewer training steps), can provide a better measurement suggestion when compared with a completely non-Bayesian approach.

