# OpenReview forum: "Deep Bayesian Experimental Design for Quantum Many-Body Systems"
_NeurIPS.cc/2023/Workshop/AI4Science — NeurIPS2023-AI4Science Poster_

### Official Review · Reviewer_Wi1E · 2023-10-24
**Review of Deep Bayesian Experimental Design for Quantum Many-Body Systems**

**Rating:** 8
**Confidence:** 3

**Review:**

**1. Summary:**

The authors explore the integration of Bayesian experimental design with deep neural networks and normalizing flows to optimize measurements for quantum technology platforms, particularly arrays of coupled cavities and qubit arrays. The paper emphasizes the value of active learning in efficiently determining the parameters of quantum systems, especially in scenarios with inherent measurement noise.

**2. Strengths:**

**(a)** The paper presents a novel approach that utilizes Bayesian experimental design and machine learning, offering some new perspectives on quantum characterization.
**(b)** The work provides insights into the challenges of characterizing quantum devices and highlights the importance of active learning and Bayesian techniques.
**(c)** The authors recognize the inherent noise in quantum measurements and provide strategies to mitigate its effects, which is often not discussed in similar works.

**3. Weaknesses:**

Clarity in Experiments: The experimental section could benefit from a more detailed presentation of the setup, datasets, and specific results to enhance the readers' understanding, especially for readers unfamiliar with quantum many-body physics.

**4. Detailed Comments:**

The paper is well-organized, especially in the methodology section, where the authors discuss the core experimental design, with a detailed discussion about the process of updating knowledge about a system based on new measurements. The Bayes rule is employed to derive the posterior distribution of system parameters after each measurement. The paper also underscores the importance of the initial prior distribution, emphasizing its role in influencing the algorithm's behavior. The discussion extends to the choice of a query function, which helps in determining the most valuable measurement to perform. A potential query function is introduced based on the expected information gain from a measurement, which relates to the mutual information between the measurement and system parameters. The authors also acknowledge the challenges in estimating the query function due to its inherent computational complexity. However, a more detailed presentation of the results in the experimental section could be helpful.

**5. Suggestions for possible Improvement:**

The reviewer believes it may be beneficial to consider the following:

**(a)** Expand the experimental section with detailed data, metrics, and visual representations to illustrate results more effectively.
**(b)** Add visualizations that help explain the quantum systems and their characterizations.
**(c)** Further exploration and detailed explanations of the methods and techniques used could be helpful.
**(d)** The paper mentions the difficulties in evaluating the expected information gain and the challenges with the nested Monte Carlo estimates. The reviewer believes providing practical solutions or alternatives to these challenges would be helpful.

**6. Reproducibility:**

The authors have discussed their methodologies and approaches. However, the availability of datasets, code, or specific implementation details hasn't been specified, making it potentially hard to comment on the reproducibility of the results.

**7. Overall Evaluation:**

The paper offers valuable insights into the challenges and solutions of characterizing quantum systems using Bayesian experimental design. The integration of Bayesian techniques with machine learning methodologies stands as a significant contribution. Despite some potential improvements in experimental details, detailed results presentation, and reproducibility, the reviewer suggests the paper be accepted in the workshop.

---

### Official Review · Reviewer_f3VU · 2023-10-24
**A new attempt for quantum experiment design using deep bayesian with normalization flow**

**Rating:** 6
**Confidence:** 4

**Review:**

Algorithm:
The paper proposes a new approach for deep bayesian learning with conditional normalization flow. It will be helpful if the authors can elaborate how this approach is compared to other bayesian or active learning methods.

Science:
The paper studied the setups of coupled cavities and an array of N qubits, which are good scenarios to be considered. The simulation results look encouraging, but it is not clear whether there are other physics or approximate methods that can also achieve similar performances. Meanwhile, it is still an open question how the proposed method works by considering the measurement complexity in real experimental scenario.

Writing:
The paper is clearly written. It reviews the current literature and methodology of the activate learning and bayesian learning. It will be very useful to provide more explanation or intuition to the quantum problem setup in main text, especially for broad readers.

---

### Meta-Review · Area_Chair_nL21 · 2023-10-27

**Recommendation:** Accept (Poster)
**Confidence:** 4

**Metareview:**

The paper combines deep Bayesian learning with conditional normalization flow.
The paper appears clearly written and is well-organized. It reviews the current literature and methodology of active and Bayesian learning.
Moreover, I believe this paper deals with a very interesting topic that would be relevant to the workshop.

Given the positive agreement between the reviewers, I recommend acceptance of the paper.

However, I strongly recommend the authors to read through and incorporate, the feedback from the authors for the camera-ready version of the paper as these changes could bring some valuable additions to the paper.